# Human iPSC Modeling of Genetic Febrile Seizure Reveals Aberrant Molecular and Physiological Features Underlying an Impaired Neuronal Activity

**DOI:** 10.3390/biomedicines10051075

**Published:** 2022-05-05

**Authors:** Stefania Scalise, Clara Zannino, Valeria Lucchino, Michela Lo Conte, Luana Scaramuzzino, Pierangelo Cifelli, Tiziano D’Andrea, Katiuscia Martinello, Sergio Fucile, Eleonora Palma, Antonio Gambardella, Gabriele Ruffolo, Giovanni Cuda, Elvira Immacolata Parrotta

**Affiliations:** 1Department of Experimental and Clinical Medicine, University Magna Graecia of Catanzaro, Viale Europa, 88100 Catanzaro, Italy; stefania.scalise@unicz.it (S.S.); clara.zannino@unicz.it (C.Z.); valeria.lucchino@unicz.it (V.L.); michela.loconte@studenti.unicz.it (M.L.C.); scaramuzzino.luana@unicz.it (L.S.); 2Department of Biotechnological and Applied Clinical Sciences (DISCAB), University of Aquila, 67100 Aquila, Italy; pierangelo.cifelli@univaq.it; 3Department of Physiology and Pharmacology, University of Rome, Sapienza, P.le Aldo Moro, 5, 00185 Rome, Italy; t.dandrea@uniroma1.it (T.D.); sergio.fucile@uniroma1.it (S.F.); eleonora.palma@uniroma1.it (E.P.); 4IRCCS Neuromed, Via Atinense, 86077 Pozzilli, Italy; katiuscia.martinello@neuromed.it; 5Department of Medical and Surgical Sciences, University Magna Graecia of Catanzaro, Viale Europa, 88100 Catanzaro, Italy; a.gambardella@unicz.it (A.G.); parrotta@unicz.it (E.I.P.); 6IRCCS San Raffaele Roma, Via della Pisana, 00163 Rome, Italy

**Keywords:** febrile seizure, induced pluripotent stem cells, mesial temporal lobe epilepsy, voltage gated sodium channel Na_V_1.1, disease model

## Abstract

Mutations in *SCN1A* gene, encoding the voltage-gated sodium channel (VGSC) Na_V_1.1, are widely recognized as a leading cause of genetic febrile seizures (FS), due to the decrease in the Na^+^ current density, mainly affecting the inhibitory neuronal transmission. Here, we generated induced pluripotent stem cells (iPSCs)-derived neurons (idNs) from a patient belonging to a genetically well-characterized Italian family, carrying the c.434T > C mutation in *SCN1A* gene (hereafter SCN1A^M145T^). A side-by-side comparison of diseased and healthy idNs revealed an overall maturation delay of SCN1A^M145T^ cells. Membranes isolated from both diseased and control idNs were injected into *Xenopus* oocytes and both GABA and AMPA currents were successfully recorded. Patch-clamp measurements on idNs revealed depolarized action potential for SCN1A^M145T^, suggesting a reduced excitability. Expression analyses of VGSCs and chloride co-transporters *NKCC1* and *KCC2* showed a cellular “dysmaturity” of mutated idNs, strengthened by the high expression of SCN3A, a more fetal-like VGSC isoform, and a high *NKCC1*/*KCC2* ratio, in mutated cells. Overall, we provide strong evidence for an intrinsic cellular immaturity, underscoring the role of mutant Na_V_1.1 in the development of FS. Furthermore, our data are strengthening previous findings obtained using transfected cells and recordings on human slices, demonstrating that diseased idNs represent a powerful tool for personalized therapy and ex vivo drug screening for human epileptic disorders.

## 1. Introduction

Febrile seizures (FS), i.e., seizures occurring during fever not due to a central nervous system (CNS) infection, are convulsive events commonly affecting children [1]. Retrospective studies have linked childhood FS to the development of hippocampal sclerosis (HS) and mesial temporal lobe epilepsy (MTLE) later in life, especially when in the presence of a family history of febrile convulsions [2]. MTLE with HS is a drug-resistant form of epilepsy and often requires patients to undergo temporal lobotomy to achieve seizure freedom [3]. Missense and nonsense mutations in the *SCN1A* gene, encoding the α-subunit of the Na_V_1.1 VGSC, are associated in a wide range of epileptic disorders in which FS are involved, such as Dravet syndrome [4], generalized epilepsy with febrile seizures plus (GEFS+) [5], simple FS [6] and MTLE with HS [7]. Additionally, SCN1A mouse mutants exposed to recurrent early-life FS developed an increased risk of seizures susceptibility during adult life [8]. Although very useful, animal models fail in trying to faithfully recapitulate the mechanisms underlying human disease, since the patient-specific genetic background is not taken into account. For instance, remarkable differences between GEFS+ [9] and simple FS [6] clinical phenotypes do exist among patients carrying the same mutation. Limitations associated with animal models can be overcome by the generation of iPSCs from a patient’s own cells [10,11,12,13]. In this study, we generated iPSCs from a patient carrying a missense mutation in the *SCN1A* gene. This mutation causes the substitution of a highly conserved methionine with a threonine in position 145 of the Na_V_1.1 protein (M145T). This patient belongs to an Italian family of 13 individuals carrying the mutation and affected by FS [6]. In addition, this patient developed MTLE with HS during adolescence, showing severe recurrent drug-resistant seizures [14,15]. Then, neurosurgery became necessary at the age of 27 years to remove hippocampal sclerotic tissue and achieve a control of seizures [15]. Patch-clamp recordings in human cell lines, transfected with a plasmid carrying the *SCN1A*-M145T mutant gene, revealed a loss-of-function mutation leading to a 60% reduction in the Na+ current density and a positive shift of about 15 mV in the voltage-dependent activation of the channel [6]. Furthermore, electrophysiological experiments conducted on fresh hippocampal slices obtained from the same patient from which iPSCs were generated, showed a more depolarized action potential (AP) threshold and an impairment of GABAergic neurotransmission in interneurons [15], a hallmark of *SCN1A* mutations in epileptic phenotypes [16,17]. Notably, the latter was coupled to an increase in GABA current use-dependent desensitization in oocytes micro-transplanted with the same hippocampal tissue [15,18].

The pivotal involvement of inhibitory interneurons in epilepsy was also shown in studies based on iPSCs models of pathogenic *SCN1A* mutations [19,20]; others have instead demonstrated the involvement of both glutamatergic and GABAergic populations in the epileptic brain hyperexcitability [21]. In this study, we used patient-specific iPSCs-derived neurons (idNs) to investigate the molecular and electrophysiological mechanisms underlying the SCN1A^M145T^ disease phenotype. Our results show a significant alteration in the development and maturation processes of SCN1A^M145T^ idNs compared to their healthy control counterpart. Electrophysiological measurements conducted on single neurons during their development add further knowledge to this scenario, with findings that successfully recapitulate those previously recorded on hippocampal sclerotic tissue from the same patient. Taken together, our results strengthen the potential of iPSCs technology for a more comprehensive understanding of the complexity of epileptic-like human phenotypes.

## 2. Materials and Methods

### 2.1. Clinical Features and iPSCs Generation from a Patient with Missense Mutation in the SCN1A Gene

In this study, we generated iPSCs from a male subject who carried a missense mutation (c.434T > C in exon 3) in the *SCN1A* gene encoding for the α-subunit of the Na_V_1.1 VGSC. The patient (referred as subject IV-3 in the pedigree described in [6,14]) belongs to a family in which thirteen members were affected by FS during childhood, all carrying the same c.434T > C missense mutation, which causes the substitution of a highly conserved methionine residue with a threonine within the S1 segment of the domain 1 in the Na_V_1.1 channel (Figure 1A). The patient of this study experienced FS lasting up to 15 min, suggesting a clinical phenotype of complex FS, until the age of six. Seven years later, he started suffering from focal complex seizures, compatible with MTLE. The disease progressed, and the patient became refractory to antiepileptic drugs (AEDs). Neurological evaluation reported bilateral mesial temporal epileptiform spikes mostly localized (>70%) on the right side, while brain MRI evidenced significant sclerosis in the right hippocampus, requiring right temporal lobectomy to achieve seizure freedom (Colosimo et al., 2007). iPSCs were also generated from skin fibroblasts isolated from a healthy thirty-year-old male and were used as a control line in our experiments. The generation and characterization of SCN1A^M145T^ and healthy control iPSCs is described in [22] (line identified as UNIMGi001-A) and [23] (see line hiPSCs-3), respectively. iPSCs were cultured on Matrigel-coated dishes with mTeSR1 medium (StemCell Technologies, Vancouver, BC, Canada), in a humidified incubator at 37 °C at 5% CO_2_. Cells were split every 4–5 days (80% confluence) with the use of Gentle Cell Dissociation Reagent (StemCell Technologies). Both cell lines were routinely tested for Mycoplasma with the Mycoplasma PCR Detection Kit (Applied Biological Materials, Richmond, BC, Canada).

### 2.2. Generation of iPSCs-Derived Neurons (idNs)

We coaxed iPSCs from both control and SCN1A^M145T^ to differentiate into neural stem cells (NSCs) using Gibco^®^ PSC Neural Induction Medium (Thermo Fisher Scientific, Waltham, MA, USA), following the manufacturer’s instructions. To obtain neurons, NSCs were then plated at a density of 5 × 10^4^ cells/cm^2^ on dishes coated with Poly-D-Lysine (molecular weight 30,000–70,000) plus Laminin (both from Merck, Darmstadt, Germany) and cultured in Neuronal Differentiation Medium (NDMC), composed of Neurobasal Medium, 1× B27 supplement, 1× Glutamax, 1× CultureOne™ Supplement, 200 μM ascorbic acid and 0.2% Penicillin/Streptomycin (all from Thermo Fisher Scientific). NDMC was supplemented with GDNF at 10 ng/mL and BDNF at 20 ng/mL (both from PeproTech, London, UK) at NSCs plating; the concentration of GDNF and BDNF was lowered to 5 ng/m and 10 ng/m, respectively, at the first medium change. Subsequently, NDMC medium was supplemented with BDNF only, used at 5 ng/mL during the second medium change and at 2.5 ng/mL during the whole culture period. Neurons were kept in culture 28–35 days until they reached full maturation for subsequent analysis.

### 2.3. RNA Extraction and qRT-PCR Analysis

Total RNA was obtained by phenol/chloroform extraction using TRIzol reagent (Thermo Fisher Scientific) and 1 μg RNA was retro-transcribed in cDNA using the High-Capacity cDNA Reverse Transcription Kit (Thermo Fisher Scientific). cDNA was used for relative quantitation of gene expression by qRT-PCR, using the SensiFAST SYBR Hi-ROX kit (Meridian Bioscience, Cincinnati, OH, USA). Gene expression levels were normalized to Glyceraldehyde 3-phosphate dehydrogenase (*GAPDH*) as a housekeeping gene. qRT-PCR was performed by QuantStudio™ 7 Pro Real-Time PCR System (Thermo Fisher Scientific). A list of primers used in this study is provided in Appendix A.

### 2.4. cDNA Sequencing

Primers based on the cDNA sequence of the *SCN1A* gene were designed to amplify the exon 3 in which the c.434T > C mutation is found. The amplification of the target region was obtained with the use of specific primer pairs (FW: 5′-ATTGAAAGACGCATTGCAGA-3′ and RV: 5′-TGTTCCTCCAAGGAAGCATT-3) and the following PCR program: 3 min at 95 °C, 30 cycles of 30 s at 95 °C, 30 s at 52 °C and 45 s at 72 °C, with a final extension at 72 °C for 5 min. Following gel electrophoresis to confirm the amplicon size (797 bp), PCR products were extracted from gel using the EZ-10 Spin Column DNA Gel Extraction Kit (Bio Basic Inc., Markham, ON, Canada ) and Sanger sequenced (Eurofins Genomics, Ebersberg, Germany).

### 2.5. Western Blot Analysis

For total protein extraction, cells were harvested in ice-cold phosphate-buffered saline (PBS) and lysed in RIPA buffer (Merck) containing Halt™ Protease Inhibitor and Halt™ Phosphatase Inhibitor Cocktails (Thermo Fisher Scientific). Protein concentration was measured by Bradford assay. After denaturation for 10 min at 70 °C in Laemmli Sample Buffer, 70 μg of proteins were resolved in acrylamide/bisacrylamide precast gels Mini-PROTEAN TGX (Bio-Rad, Hercules, CA, USA) and transferred to nitrocellulose membrane. The membrane was incubated overnight at 4 °C with the following primary antibodies: anti-Na_V_1.1 (rabbit polyclonal, 0.5 μg/mL, ab24820, Abcam, Cambridge, UK) and anti-TUBB3 (mouse monoclonal, 1:10,000, 480,011, Thermo Fisher Scientific). After washing, horseradish peroxidase conjugated secondary antibody anti-rabbit IgG and anti-mouse IgG (Jackson ImmunoResearch, Cambridge, UK) were added to the membrane and incubated for 1 h at room temperature. The protein bands on the membranes were detected by Clarity™ Western ECL Blotting Substrates (Bio-Rad) using the Alliance™ Q9-Atom (Uvitec, Cambridge, UK). Western blot bands were quantified using the Analyze Gels tool of Fiji Software [24].

### 2.6. Immunofluorescence

Immunofluorescence analysis was performed on poly-D-Lysine plus laminin-coated permanox chamber slides (Thermo Fisher Scientific). Cells were fixed in 4% (vol/vol) paraformaldehyde (PFA) and subjected to immunostaining with the following primary antibodies: anti-TUBB3 (mouse monoclonal, 1:250, 480011, Thermo Fisher Scientific), anti-MAP2 (mouse monoclonal, 1:1000, MA5-12826, and chicken polyclonal, 1:5000, PA1-10005, both from Thermo Fisher Scientific), anti-NEFH (rabbit polyclonal, 1:1000, ab8135, Abcam), anti-GAD1 (chicken polyclonal, 1:1000, AP31805PU-N, Origene, Rockville, MD, USA), anti-SST (mouse monoclonal, 1:200, Ma5-17182, Thermo Fisher Scientific), anti-CALB2 (rabbit polyclonal, 1:100, PA5-16681, Thermo Fisher Scientific) and anti-CALB1 (Rabbit monoclonal, 1:100, NB120-11427, Abcam), anti-Na_V_1.1 (rabbit polyclonal, 1:100, ab24820 Abcam) and anti-vGLUT1 (mouse monoclonal, 1:100, sc-377425, Santa Cruz Biotechnology, Dallas, TX, USA). Incubation with primary antibodies was carried overnight at 4 °C. After washing with PBS, cells were incubated with AlexaFluor-594, or -488 conjugated secondary antibodies (all from Thermo Fisher Scientific) for 1 h at room temperature. Nuclei were stained with DAPI (4′,6-diamidino-2-phenylindole, Thermo Fisher Scientific) and mounted with Dako Fluorescent Mounting Medium (Agilent, Santa Clara, CA, USA). Images were acquired with a Leica microscopy system (DMi8), using LAS X (v.3.7.4.23463) software. For quantification of double positive cells (Figure 2D and Figure Figure 3D,E,F), neurons from two different differentiation experiments were manually counted using the multipoint tool in Fiji software in a blinded manner.

### 2.7. Patch-Clamp Recordings on idNs

Whole-cell patch clamp recordings were performed on idNs of WT and SCN1A^M145T^ mutant at day of differentiation 35 in 35 mm Petri-dishes. The measures were performed at 25 °C. The identification of neurons followed morphological criteria: highly birefringent cells with small diameter processes were selected, and 100% of the WT cells exhibited action potentials (APs). APs were recorded from neurons applying depolarizing current steps (4–50 pA, 1 s) using glass electrodes (3–4 MΩ) filled with (in mM): 140 KCl, 10 Hepes, 5 BAPTA, 2 Mg-ATP (pH 7.3, adjusted with KOH). Membrane potentials were acquired at 50 kHz and filtered at 3 kHz with an amplifier HEKA EPC 800 (HEKA Elektronik, Reutlingen, Germany) and analyzed off-line. During recordings, cells were continuously perfused using a gravity-driven perfusion system with the following external solution: 140 mM NaCl, 10 mM HEPES, 2.8 mM KCl, 2 mM CaCl_2_, 2 mM MgCl_2_, 10 mM glucose, (pH 7.3 adjusted with NaOH). Membrane potentials have been corrected for junction potential.

### 2.8. Membrane Preparation from idNs

We isolated cellular membranes from approximately 7 × 10^7^ idNs of WT and SCN1A^M145T^. The procedure was similar to that already described in [18] for human tissues. The cells were scraped at day of differentiation 35, spun down and subsequently homogenized in membrane buffer (200 mM glycine, 150 mM NaCl, 50 mM EGTA, 50 mM EDTA, and 300 mM sucrose; plus 10 μL/mL of protease inhibitors, P2714 (Sigma)—pH 9 adjusted with NaOH). Then, the vials were centrifuged for 15 min at 9500× *g*. Afterwards, the supernatant was centrifuged for 2 h at 100,000× *g* with an ultracentrifuge. Finally, the pellet was resuspended in glycine 5 mM and used directly or aliquoted and kept at −80 °C for later usage.

### 2.9. Xenopus Laevis Oocytes Injection and Voltage-Clamp Recordings

*Xenopus* oocytes were collected and injected as previously described in [18]. The animal protocols were approved by the Italian Ministry of Health (authorization no. 427/2020-PR). The electrophysiology experiments were carried out from 12 to 48 h after injection, with the technique of two-electrode voltage-clamp. The two microelectrodes were filled with 3M KCl. The oocytes were placed in a recording chamber (0.1 mL volume) and perfused continuously with oocyte Ringer solution (OR: NaCl 82.5 mM; KCl 2.5 mM; CaCl_2_ 2.5 mM; MgCl_2_ 1 mM; Hepes 5 mM, adjusted to pH 7.4 with NaOH) at room temperature (20–22 °C). The neurotransmitters (GABA or AMPA) were administered through a gravity driven multi-valve perfusion system (9–10 mL/min) controlled by a computer (Biologique RSC-200; Claix, France) to ensure the exact duration of each application. AMPA currents were recorded in presence of cyclothiazide (CTZ, 20 μM) in order to avoid receptor desensitization [18]. GABA, AMPA, CTZ, Bicuculline methochloride and NBQX were purchased from Tocris Bioscience (Bristol, UK) and dissolved in sterile water (GABA, AMPA and Bicuculline methochloride) or DMSO (CTZ, NBQX) before final dilution to the desired concentration in OR. The solutions containing DMSO were always used with a final DMSO concentration lower than 1:1000. GABA current rundown was defined as the decrease in the current peak amplitude after six 10 s applications of GABA at 40 s intervals, expressed as percentage of the first response [25].

### 2.10. Statistical Analysis

For molecular biology data, the number of biological replicates used in each experiment was indicated in the figure legends. Statistical analysis was performed using two-tailed *t*-test or multiple unpaired *t*-test with Welch correction in GraphPad Prism software, version 9.3.1. Data are represented as means of biological replicates ± SEM and *p*-values ≤ 0.05 were considered significant. For patch-clamp experiments, data sampling and analysis were performed using pClamp 10 software (Molecular devices, Sunnyvale, CA, USA). Statistical significance was assessed with ANOVA, unless otherwise stated.

The figures of this work were created with BioRender.com.

## 3. Results

### 3.1. Generation of SCN1A^M145T^ and Control idNs

*SCN1A* loss-of-function mutations are reported to affect both GABAergic and glutamatergic neurons [21]. Therefore, we directed WT and SCN1A^M145T^ iPSCs differentiation toward forebrain neurons. idNs presented a well-defined neuronal morphology (Figure 1B) and a high expression of neuronal marker genes *MAP2*, *NEFL*, *NEFM*, *SYP*, and *PSD95* with respect to their undifferentiated counterpart (iPSCs) (Figure 1C), while *ALDH1L1* and *OLIG2,* astrocytes- and oligodendrocytes-specific markers, respectively, are expressed at lower level compared to *MAP2* expression (Appendix A). Moreover, immunofluorescence analysis showed that idNs co-express the pan-neuronal marker protein TUBB3, dendrite marker MAP2, and axonal marker NEFH (Figure 1D).

**Figure 1 biomedicines-10-01075-f001:**
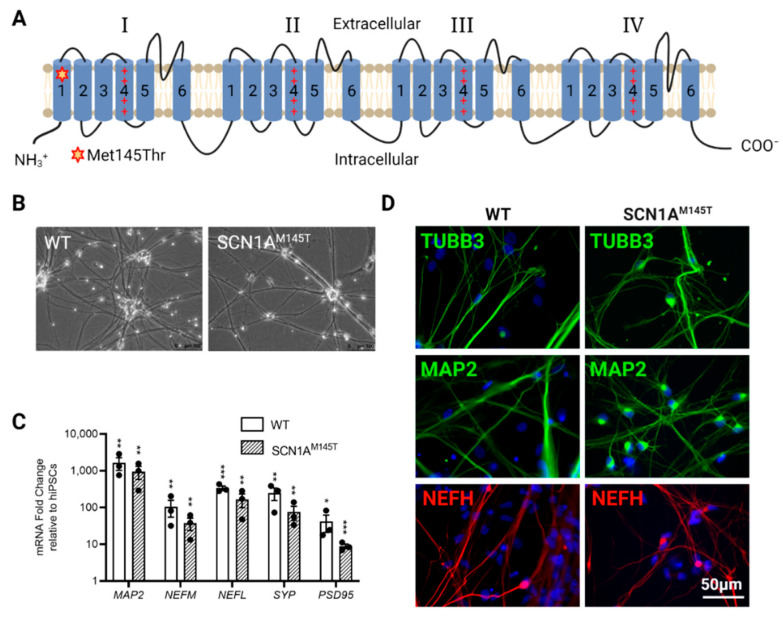
Characterization of idNs. (**A**) Representation of Na_V_1.1 channel. The star in segment 1 of domain I shows the localization of the mutated aminoacid (Met145Thr). The relative missense mutation c.434T > C is found in the exon 3 of the translated sequence. (**B**) Bright-field images of idNs from WT and SCN1A^M145T^-iPSCs (20× magnification). (**C**) Differentiated idNs show high expression levels of neuronal specific genes such as *MAP2*, *NEFM*, *NEFL, SYP* and *PSD95* compared to their undifferentiated counterparts (iPSCs). *GAPDH* was used as a housekeeping control. Data are presented as mean ± SEM of three biological replicates (black dots), * *p* < 0.05, ** *p* < 0.01, *** *p* < 0.001, *t*-test has been calculated vs. expression in iPSCs. (**D**) Immunostaining of neuronal markers TUBB3 (neurites marker), MAP2 (cell body and dendrites marker), and NEFH (axonal marker) in WT and SCN1A^M145T^ idNs. DAPI nuclear counterstain is shown in all images in blue (63× magnification).

Morphological analysis of cultured idNs revealed the presence of both bipolar (inhibitory) and pyramidal (excitatory) populations (Figure 2A). The expression of glutamate decarboxylase (*GAD2*, GABAergic marker) and vesicular glutamate transporter (v*GLUT2*, glutamatergic marker) by qRT-PCR analysis revealed a slightly higher expression of the GABAergic marker in idNs from WT and SCN1A^M145T^ (Figure 2B). In addition, by immunofluorescence analysis, we found that about 90% of MAP2^+^ neurons indeed co-express the GABAergic marker GAD1 (Figure 2C,D). On the other hand, the glutamatergic marker vGLUT1 could be slightly detected in few cells only (Appendix A).

**Figure 2 biomedicines-10-01075-f002:**
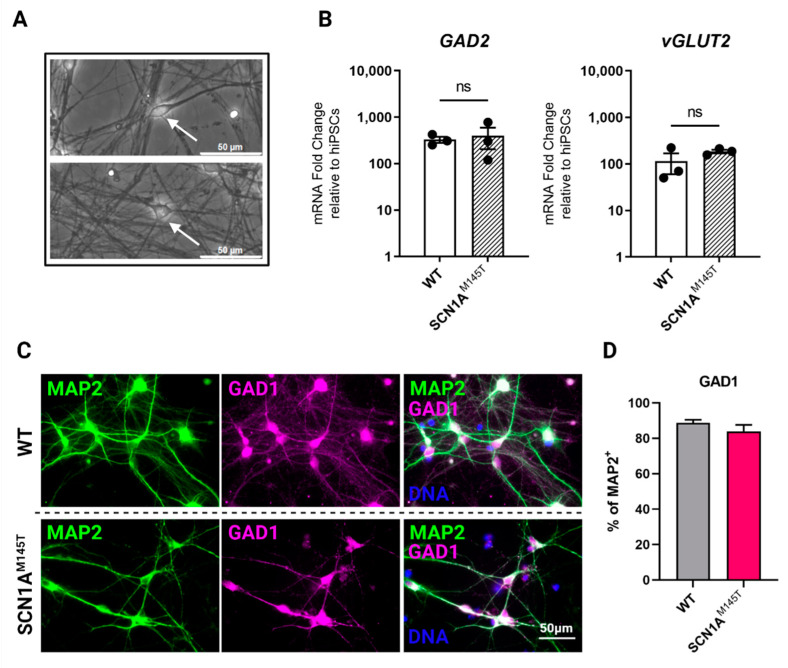
Expression of GABAergic and glutamatergic markers in idNs. (**A**) Generated idNs are composed of a mixed neuronal population containing neurons with both bipolar (upper panel, white arrow) and pyramidal (lower panel, white arrow) morphology. (**B**) Expression analysis of mRNAs relative to GABAergic (inhibitory) marker glutamate decarboxylase 2 (*GAD2*) and glutamatergic (excitatory) marker vesicular glutamate transporter 2 (*vGLUT2*) in idNs relative to undifferentiated iPSCs. *GAPDH* was used as a housekeeping gene. qRT-PCR analysis did not reveal significant differences in the expression of *GAD2* and *vGLUT2* between diseased and control idNs relatively to their undifferentiated iPSCs (*p*-value = non-significant (ns), *t*-test), even though a higher prevalence of GABAergic marker *GAD2* was detected, as shown by its higher fold change relative to iPSCs. Data are presented as mean ± SEM of three biological replicates (dots). (**C**) Representative immunofluorescence images of GABA synthesis enzyme GAD1 compared to neuronal marker MAP2 in idNs of WT (upper line images) and SCN1A^M145T^ (lower line images) cells (63× magnification). (**D**) The diagram shows that about 90% of MAP2^+^ cells co-express the GAD1 marker. For each cell line, at least 300 neurons were counted, and data are presented as mean ± SEM of two independent experiments.

Mature GABAergic interneurons express specific neuropeptides and calcium binding protein, thus we performed immunostaining of idNs at day 35 of differentiation using antibodies against somatostatin (SST) (Figure 3A), Calretinin (CALB2) (Figure 3B), Calbindin (CALB1) (Figure 3C), and parvalbumin (PV). Results of immunofluorescence analysis indicate that 10% of MAP2^+^ neurons expressed SST (Figure 3D and Appendix A), about 9% expressed CALB2 (Figure 3D and Appendix A), while CALB1 is slightly present (less than 1% of MAP2^+^, Figure 3D and Appendix A). We could not detect PV positive neurons in idNs from WT and SCN1A^M145T^ in line with the observation that this protein is expressed later during neuronal differentiation of iPSCs [26].

**Figure 3 biomedicines-10-01075-f003:**
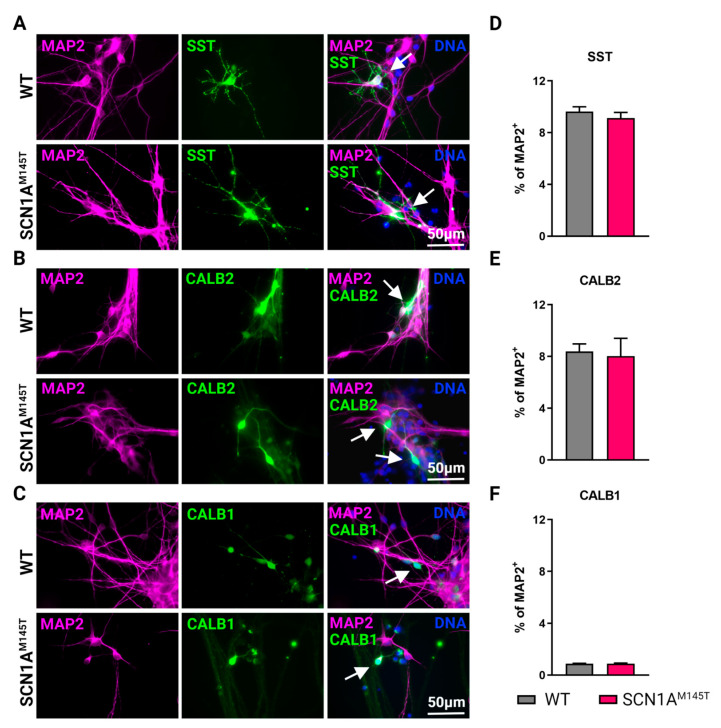
Types of interneurons generated from iPSCs. Immunofluorescence analysis of idNs stained with antibodies against interneuronal subtypes markers (**A**) somatostatin (SST), (**B**) calretinin (CALB2), and (**C**) calbindin (CALB1). In each group of images, WT cells are shown in the upper panel, while SCN1A^M145T^ idNs are shown in the lower panel. White arrows in the merged images indicate neurons expressing the interneuronal makers indicated (63× magnification). (**D***–***F**) Quantification of percentage of MAP2^+^ neurons co-expressing interneuronal markers immunostained in panels (**A**–**C**). About 9–1% of idNs express SST and CALB2, while CALB1 is present in less than 1% percent of neurons. At least 200 cells were counted for each bar, and data are presented as mean ± SEM of two independent experiments.

### 3.2. Expression of SCN1A Gene and Na_V_1.1 Protein in idNs

Given that the SCN1A^M145T^ patient is heterozygous for the mutation, we inquired whether both *SCN1A* alleles are transcribed in idNs. To this purpose, we used Sanger sequencing to analyze cDNA obtained by reverse transcription of *SCN1A* mRNA derived from idNs of both WT and the SCN1A^M145T^. Interestingly, the results confirmed the heterozygous status in SCN1A^M145T^-idNs, as demonstrated by the presence of a double peak at the mutation site (Figure 4A). Subsequently, we performed a side-by-side analysis of the expression of *SCN1A* mRNA together with others VGSCs known to be predominantly expressed in CNS, such as *SCN2A*, *SCN3A* and *SCN8A*, in idNs at different time points during differentiation (d0, which corresponds to the NSCs stage, d14, d21, and d28). Interestingly, we found that, although the expression of *SCN1A* increased in both WT- and SCN1A^M145T^ idNs over time in culture, the expression of *SCN1A* channel results lower, in all time-points tested, in patient idNs compared to control idNs (Figure 4B). We observed a similar trend of expression also for the *SCN2A,* which has the higher expression level among all the sodium channels analyzed (Appendix A). On the other hand, *SCN3A*, which is considered an embryonic isoform [27], progressively decreases during the differentiation period in WT idNs, while mutated idNs show an opposite expression trend of *SCN3A,* which results up-regulated over time in culture (Figure 4C). Lastly, the expression of *SCN8A* channel was low in both control and mutated idNs (Appendix A), in accordance with the notion that *SCN8A* is poorly expressed in developing neurons [27]. Based on the difference found in the expression of *SCN1A* at the mRNA level, we analyzed the expression levels of Na_V_1.1 protein in total lysates from control and mutated idNs. Intriguingly, immunoblot analysis showed that SCN1A^M145T^ neurons express significantly lower levels of Na_V_1.1 protein compared to healthy neurons (Figure 4D,E). Moreover, our immunofluorescence data revealed a high expression of Na_V_1.1 within the soma of GAD1^+^ neuronal cells (Figure 4F), in line with the in vivo data showing that Na_V_1.1 is primarily expressed in GABAergic neurons [17].

### 3.3. Expression of Chloride Cotransporters in idNs

The functionality of GABA neurons during development is intimately correlated with the expression of the chloride co-transporters, the Na-K-2Cl cotransporter isoform 1 (NKCC1) and the K-Cl cotransporter isoform 2 (KCC2) [28] and a high NKCC1/KCC2 ratio indicates neuronal immaturity [29,30]. We analyzed the expression of the two chloride co-transporters transcripts in idNs from WT and SCN1A^M145T^ at day 0 of differentiation (neural stem cells), d14, d21, d28, and d60. In accordance with data reported in the human brain transcriptome database [31], we observed that *KCC2* undergoes strong physiological increase in idNs from WT during development, while its expression remained at low levels in idNs from SCN1A^M145T^ during the whole experimental culture period (Figure 5A). The expression of *NKCC1* was instead similar between the two groups during the first phases of neuronal development (from d0 to d28) but became significantly over-expressed at day 60 of differentiation in SCN1A^M145T^ idNs only (Figure 5B). Additionally, the *NKCC1*/*KCC2* mRNA expression ratio was higher in mutated neurons at all differentiation time points tested (Figure 5C). Altogether, these findings provide strong evidence that an imbalanced NKCC1/KCC2 expression shift occurs in idNs derived from the patient carrying the M145T mutation in the *SCN1A* gene, suggesting that this mutation may promote the persistence of an immature phenotype.

### 3.4. Recording of GABA and AMPA Currents by Injection of idNs Membranes in Xenopus Oocytes

Here, we recorded for the first time neurotransmitter-evoked currents from *Xenopus* oocytes injected with membranes obtained from idNs (Figure 6). First, we successfully evoked GABA and AMPA responses both from SCN1A^M145T^-injected oocytes and control (WT)-injected oocytes. We obtained responses that ranged from to 3.1 nA to 75.0 nA for GABA (500 μM, 4 s, mean = 18.2 ± 2.5 nA; *N* = 44) and from 6.3 to 43.2 nA to for AMPA (20 μM, 10 s with a short 20 s pretreatment with CTZ 20 μM, mean = 16.4 ± 1.6; n = 27). In order to verify that currents we recorded were genuine, we blocked the evoked current with the respective specific blockers (Figure 6A,B). As expected, GABA currents (500 μM) were totally blocked by co-application of bicuculline, a competitive antagonist of GABA_A_Rs, (100 μM; n = 6) and AMPA currents (20 μM) were totally blocked by the specific AMPA receptor blocker NBQX, a competitive antagonist of AMPARs (50 μM; n = 6). Both GABA (Figure 6A) and AMPA (Figure 6B) currents recovered their original amplitude after the washout of the blocker. In another set of experiments, we measured the GABA current rundown, a GABAergic dysfunction associated with drug-resistant epilepsy [25,32,33], from SCN1A^M145T^-injected oocytes and WT-injected oocytes. Not surprisingly, we measured a value of current rundown in WT-injected oocytes that was similar to that recorded in previous studies [34] using cortical tissue samples from individuals without any neurological disorder (71.0 ± 2.7%; n = 8; Figure 6C). On the other hand, interestingly, we did not measure a significant increase in current rundown in SCN1A^M145T^-injected oocytes (65.7 ± 3.2%; n = 10; Figure 6C).

### 3.5. Patch-clamp recordings of WT and SCN1A^M145T^ idNs

WT- and SCN1A^M145T^ idNs at day of differentiation 35 were analyzed to compare their functional properties. Resting membrane potential and cell capacitance values were similar in WT and SCN1A^M145T^: −48 ± 2 mV vs. −44 ± 2 mV, and 28 ± 3 pF vs. 35 ± 2 pF, respectively. All WT neurons tested fired APs upon current injection (Figure 7A, 15 out 15 cells), while responsive SCN1A^M145T^ idNs were the 62% of the total (Figure 7A inset, 20 out 34 cells; *p* = 0.004, Fisher Exact test). Although at this stage AP threshold is not expected to be at the full maturation level, WT neurons exhibited a more hyperpolarized mean AP threshold value than SCN1A^M145T^ (−37 ± 1 mV vs. −31 ± 1 mV, *p* = 0.003; Figure 7A–C, [35]). Furthermore, AP amplitudes were larger in WT than in SCN1A^M145T^ idNs (57 ± 3 mV vs. 45 ± 2 mV, *p* = 0.006; Figure 7D), while no differences were observed in AP kinetics.

## 4. Discussion

Na_V_1.1 sodium channel, encoded by *SCN1A* gene, belongs to the family of VGSCs and allows the sodium influx from extracellular space into the cytosol during depolarization. Na_V_1.1 is highly expressed in the CNS and peripheral nervous system (PNS) and mainly localizes in the cell bodies and proximal processes of neurons, where it is involved in the generation of action potential [36,37]. Mutations in the *SCN1A* gene are responsible for a plethora of diseases, collectively known as channelopathies affecting the entire nervous system [38,39]. In particular, pathogenic variants in *SCN1A*/Na_V_1.1 are responsible for several epilepsy syndromes including Dravet Syndrome (DS), a severe childhood form of epilepsy characterized by beginning with complex FS. Genetically caused FS are associated with missense loss-of-function mutations in *SCN1A*. These mutations are often functionally linked to hypoexcitability of at least some type of γ-aminobutyric acid (GABA)ergic neurons, due to decrease in Na^+^ current density [16,40,41,42,43], even if the exact mechanisms responsible for the disease are still unknown. Given that, additional approaches are necessary to unravel further physiopathological features that can be targeted by novel therapeutic strategies. In this study, we took advantage of the iPSCs technology to investigate the function underlying the clinical phenotype of a patient belonging to a well characterized Italian family with FS due to genetic defect in *SCN1A* gene [6]. Particularly, the patient described here harbors a c.434T > C missense mutation in the *SCN1A* gene (SCN1A^M145T^), responsible for the substitution of a highly conserved methionine residue with a threonine within the S1 segment of the domain 1 in the Na_V_1.1. Clinically, the patient was characterized by a complex pathophysiology with FS lasting up to 15 min [14] and developed MTLE with HS which in the end required neurosurgery. A similar condition affects a significant number of people suffering from drug-resistant epileptic seizures and, notwithstanding the recent advances that constantly improve the outcomes of surgical interventions [44], this invasive procedure is not yet completely free from complications [45]. This is mostly due to the failure of the available ASM which may effectively decrease frequency and severity of seizures without tackling the pathophysiological mechanisms, still partly unknown, that underlie their generation and recurrence [46]. It is for this reason that the search for new models and research approaches is currently one of the main topics in the field of drug-resistant epilepsy [47,48], since it may open new perspectives towards alternative therapeutic strategies.

Here, our main purpose was to build a comprehensive model of the disease by performing a side-by-side comparison of neurons differentiated from diseased (SCN1A^M145T^ idNs) and healthy control iPSCs (WT idNs). Our study allowed us to draw the following major conclusions: (i) SCN1A^M145T^ idNs show an overall immature phenotype, as demonstrated by the altered expression of the chloride co-transporters, *NKCC1* and *KCC2*, and VGSC isoforms; (ii) SCN1A^M145T^ idNs show a depolarized action potential threshold compared to the WT counterpart measured by patch-clamp, suggesting a reduced excitability; (iii) we were able, for the first time to our knowledge, to record AMPA and GABA currents from both SCN1A^M145T^- and control-idNs membranes micro-transplanted into *Xenopus* oocytes. Concerning chloride co-transporters, we could detect an increase in *NKCC1*/*KCC2* ratio for SCN1A^M145T^ idNs, that well fits with an overall immaturity of diseased neurons [49]. Furthermore, this observation is in line with our previous study of a patient affected by Dravet syndrome [50]. Indeed, previous studies have reported that GABA_A_ receptors (Rs) function is strongly dependent on chloride homeostasis ensured by the chloride co-transporters NKCC1 and KCC2 both in physiological and pathological conditions [49]. The action of NKCC1 was shown to prevail during the first phases of neuronal development, where it is involved in the depolarizing, or “less hyperpolarizing”, current through GABA_A_Rs [29] and later during development this equilibrium is shifted in favor of KCC2 [51]. Interestingly, our findings indicate that SCN1A^M145T^ shows a persistent increase in NKCC1/KCC2 mRNA ratio, which supports the hypothesis that other physiopathological mechanisms, beyond the complexity of sodium channel mutations, deserve further investigation for a comprehensive understanding of channelopathies complexity [52].

Additionally, we observed a high expression of *SCN3A*, the fetal isoform of VGSCs slightly detectable in mature cells [53], and a lower expression of *SCN1A* and *SCN2A* in SCN1A^M145T^ idNs over time in culture, while WT idNs display a VGSCs expression pattern that mirrors the changes observed during normal developmental and maturation processes [27]. The progressive up-regulation of *SCN3A* in diseased neurons may reflect a sort of compensatory effect, as previously reported in mice carrying loss-of-function mutations in *SCN1A* and where the increased Na_V_1.3 expression was observed [16]. Overall, our findings, using the most common glial and neuronal markers, are in accordance with those by other authors providing evidence that differentiation of pluripotent stem cells mainly produce interneurons [21,54]. In addition, we found that SCN1A^M145T^ neurons express significantly lower levels of Na_V_1.1 protein compared to healthy neurons, even if this protein co-localizes with GAD1. Therefore, we may hypothesize that the loss of function of the mutated protein expressed on interneurons is contributing to their decreased excitability, leading to a reduced GABA release on synaptic targets. This “interneurons hypothesis” [55] contributes, at least in part, to defining a pathophysiological substrate for the generation and recurrence of seizures in these patients [15,50]. Indeed, to further support this hypothesis, here we found a depolarized action potential threshold in SCN1A^M145T^ neurons compared to WT, although in both situations we could not record a fully developed AP threshold value as expected in these experimental conditions [35]. Interestingly, the patient from which we differentiated idNs was also suffering from drug-resistant MTLE, which prompted us to measure GABA current rundown, a GABAergic dysfunction which is a hallmark of this condition [25,32,56,57], in idNs membranes-injected oocytes. Indeed, we could not measure a significant increase in current rundown in SCN1A^M145T^-injected oocytes, in contrast to what was observed measuring this electrophysiological parameter from surgically resected brain tissue of patients with drug-resistant MTLE, including the patient of this study [15,34]. This is an important point since functional impairment of GABAergic neurons and GABA current-rundown tightly correlate with MTLE phenotype. A reasonable explanation for this discrepancy could lie in the fact that GABA rundown might arise as a part of the cascade of pathological events eventually leading to generation and recurrence of seizures [58,59]. As such, it is unlikely to detect a high GABA rundown in iPSCs-derived neurons, since these cells have never undergone continuous insults such as seizures or hippocampal sclerosis occurrence. This result offers an additional and intriguing perspective for the analysis of our results. Indeed, previous studies support the idea that GABA current rundown emerges in the “chronic” stages of the epileptic disorder, after the first spontaneous seizure [56,58]. Unfortunately, this means that there would be scarce therapeutic opportunity to prevent the consolidation of this aberration of GABAergic synaptic transmission since patients usually require medical attention after the appearance of spontaneous seizures [60]. On the other hand, there are patients that clearly carry additional “risk factors” for developing epilepsy, such as genetic mutations. Here and in previous studies [61], we clearly hypothesize that the aforementioned channelopathies can induce other synaptic dysfunctions [50,62], thus an early therapeutic intervention may be possible in conditions where a definite risk factor can be identified. Moreover, we can hypothesize that preventing the appearance of synaptic dysfunctions may have an impact on the evolution of the disease [63,64]. Clearly, additional experiments will be designed to further develop this hypothesis. For instance, future studies using the methodologies described here will allow the evaluation of the effect of candidate pharmacological agents on key pathophysiological alterations, such as those reported above. Moreover, an interesting outlook would be the implementation of our methodology with innovative and dynamic techniques of cell culture [65,66,67,68]. Additional future investigations will focus on transcriptomic and proteomic profiling for a comprehensive understanding of how genes and proteins are expressed and interconnected in the complex disease phenotype. Overall, our results, albeit limited by the fact that the data are obtained from cells generated from a single patient, are characterized by a high robustness and contribute by shedding light on the molecular mechanisms responsible for this particular form of FS, opening new stimulating perspectives on the ex vivo precision medicine approaches for a better management of patients with FS and MTLE, and for the prevention of potential development of drug resistance.

## 5. Conclusions

We report the generation and characterization of idNs from a patient belonging to a genetically well-characterized Italian family, carrying the c.434T > C mutation in SCN1A gene, responsible for FS and MTLE. Notably, electrophysiological experiments mirror the profile recorded from hippocampal tissue resected from the same patient, strengthening the validity of iPSCs technology for disease modeling. Moreover, our functional data clearly show that this channelopathy induces additional synaptic dysfunctions that may be a consequence of seizures or hippocampal sclerosis which may be prevented by early and targeted interventions. Using a multidisciplinary approach, our results reveal an aberrant maturation and altered electrophysiological features in neurons derived from the SCN1A^M145T^ patient and set the ground for future use of this approach for personalized medicine.

## Figures and Tables

**Figure 4 biomedicines-10-01075-f004:**
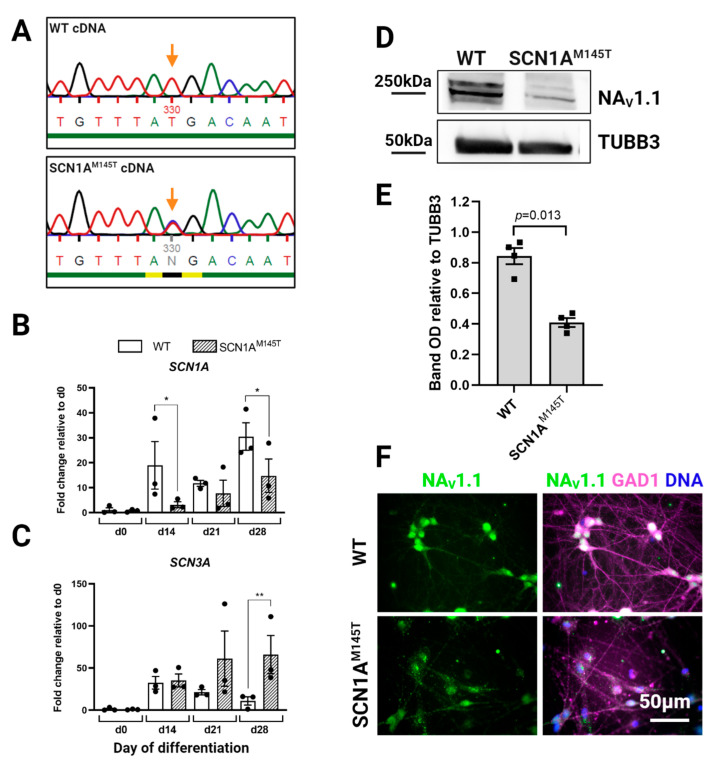
Expression of *SCN1A* gene and Na_V_1.1 protein in idNs. (**A**) Sequencing of cDNA obtained by reverse transcription of *SCN1A* mRNA from WT- and SCN1A^M145T^-idNs. In the SCN1A^M145T^ cells, the double peak in the mutation site (indicated by the orange arrow) demonstrates that both alleles (one with the original nucleotide T and the other with the mutated one C) were transcribed. (**B**) qRT-PCR analysis of *SCN1A* gene in idNs at day of differentiation 0 (NSCs), d14, d21 and d28. SCN1A^M145T^ cells showed a lower expression of SCN1A during differentiation compared to WT, although in both cell lines the expression increases following idNs maturation. (**C**) The expression of the embryonic isoform of sodium channel *SCN3A* increases with the progress of cell maturation in SCN1A^M145T^ idNs, while it decreases in WT idNs as the cells become more differentiated. For both graphs, *GAPDH* was used as a housekeeping gene; data are mean ± SEM of three biological replicates (black dots), * *p* < 0.05, ** *p* < 0.01, *t*-test has been calculated vs. WT at the same day of differentiation. (**D**) Western blot analysis of Na_V_1.1 protein in lysates obtained from WT- and SCN1A^M145T^ idNs at day 35 of differentiation. Tubulin Beta 3 Class III (TUBB3) was used as loading control. (**E**) Quantification of Na_V_1.1 Western blot bands in four biological replicates (*n* = 4, OD = relative optical density calculated as (Na_V_1.1 optical density)/(TUBB3 optical density), *p*-value calculated using *t*-test). (**F**) Immunofluorescence of idNs showing that GABAergic neurons (GAD1 positive) express Na_V_1.1, mainly in the cell body. Immunofluorescence data show a lower expression of the channel in the neurons differentiated from SCN1A^M145T^ patient in respect to those of WT (63× magnification).

**Figure 5 biomedicines-10-01075-f005:**
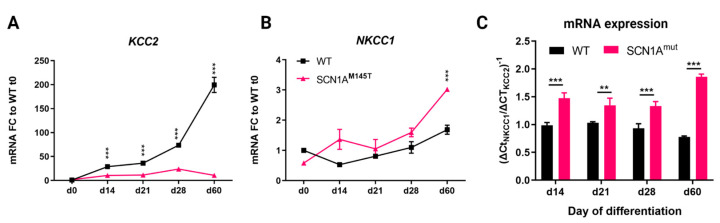
Expression of chloride co-transporters in idNs. (**A**,**B**) qRT-PCR analysis of chloride co-transporters *KCC2* and *NKCC1* in neurons from WT and SCN1A^M145T^ tested at day of differentiation 0 (NSCs), d14, d21, d28 and d60. (**C**) *NKCC1*/*KCC2* mRNA ratio in idNs at different days of differentiation. The ratio was calculated as the inverse of ΔCt_NKCC1_/ΔCt_KCC2_: (ΔCt = Ct_Gene_Of_Interest_ − Ct_GAPDH_). For all graphs, the mean ± SEM of three biological replicates is shown; ** *p* < 0.01, *** *p* < 0.001, *t*-test has been calculated vs. WT at the same day of differentiation.

**Figure 6 biomedicines-10-01075-f006:**
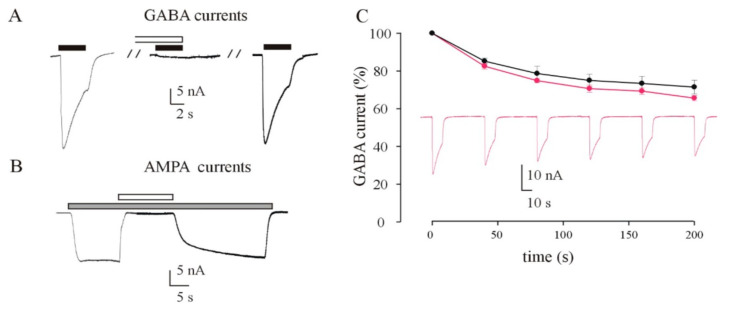
*Xenopus* oocytes injected with membranes from idNs incorporated functional neurotransmitter receptors. (**A**) Sample currents evoked by 500 μM GABA or (**B**) 20 μM AMPA on oocytes microinjected with membranes extracted from cultured idNs obtained from a patient carrying the M145T mutation of the *SCN1A* gene. (**A**) GABA currents were completely inhibited by a brief pre-incubation (30 s) with bicuculline (100 μM) and subsequently recovered following the washout of the inhibitor. (**B**) AMPA currents were completely inhibited by co-administration of NBQX (50 μM), and they recovered to the original amplitude once NBQX administration was interrupted. AMPA currents were recorded in presence of CTZ (20 μM). Black bars = GABA; gray bars = AMPA; white bars in (**A**) bicuculline; in (**B**) NBQX. (**C**) Time course of the GABA current rundown evoked by six consecutive GABA applications (500 μM, 10 s) interspaced by a 40 s washout, in oocytes injected with membranes from control (black dots; ●) and M145T idNs (magenta; ●; *p* > 0.05). The dots represent GABA currents expressed as a percentage of the first evoked response (● = 16.6 ± 1.0 nA, n = 8; ● = 23.7 ± 1.1 nA, n = 10).

**Figure 7 biomedicines-10-01075-f007:**
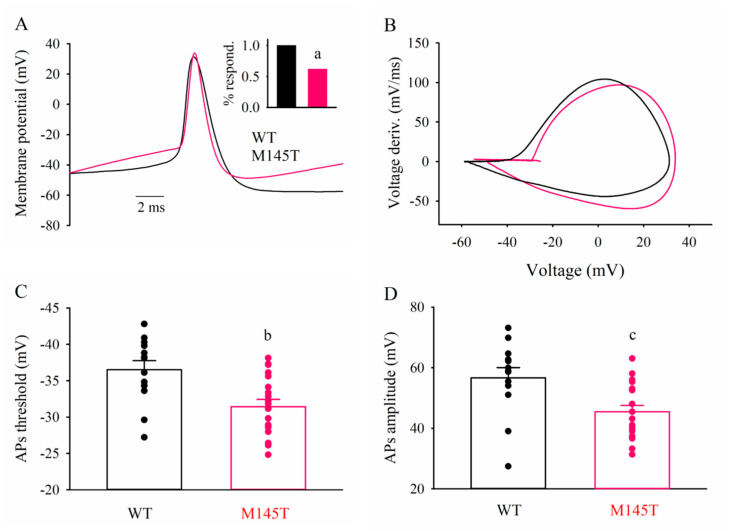
SCN1A^M145T^ idNs exhibit depolarized action potential (AP) threshold. (**A**) superimposed typical AP traces recorded from one WT and one M145T neuron (black and magenta traces, respectively). Inset: bar graphs representing the frequency of AP-firing cells (a, *p* = 0.004, Fisher Exact test). (**B**) superimposed phase-plane plot obtained from the two APs shown in (**A**). (**C**) bar graphs representing the mean AP threshold values averaged from 13 WT- and 19 SCN1A^M145T^ idNs, as indicated. Circles indicate the AP threshold of individual cells (b, *p* = 0.003). (**D**) bar graphs representing the mean AP amplitude values. Same cells as in (**C**) (c, *p* = 0.006).

## Data Availability

All the data presented in this study are available from the corresponding author upon reasonable request.

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
