# Peer review of "Human iPSC Modeling of Genetic Febrile Seizure Reveals Aberrant Molecular and Physiological Features Underlying an Impaired Neuronal Activity"

_biomedicines, 2022, doi:10.3390/biomedicines10051075_

Round 1

Reviewer 1 Report

The authors present an interesting study examining the influence of a particular mutations in the SCN1A gene in the context of incidence of seizures. Specifically, the authors utilise samples obtained from an individual presenting with the c.434T>C mutation to produce induced neuronal cultures expressing the mutation for in vitro modelling. Thereafter, with the use of a control cell type obtained from a healthy individual, the authors profile the characteristics of the ‘disease’-type neurons; measuring the gene profile of particular targets, the expression and localisation of pertinent proteins, and the measurement of action potentials, to name but a few. Together, the authors demonstrate what differences in the ‘disease’ phenotype may lead to increased incidence of seizures, and furthermore, highlight the potential of this approach to complement existing disease modelling approaches in developing therapies and interventions.

In reviewing the manuscript however, I noted a few things that the authors should assess. The following should be considered when preparing a suitable revision.

  1. The manuscript utilises microscopy on a number of occasions in assessing the expression of particular markers, with estimates provided on the co-expression of markers relative to one another in the samples examined. These data are illustrated in the form of bar graphs (e.g. Figure 2D). It is difficult to give these data weight without knowing more on the sampling strategy used to quantify this. The figure legend suggests 300 counts were performed or 200 counts depending, but how were the fields selected? Why weren’t several preparations prepared such that the consistency of the result was examined? It appears this was performed once owing to the lack of error bars. The authors must provide additional information on these data sets, particularly with regards to the sampling strategy.
  2. There are a number of figures throughout which do no appear to have any statistical analyses performed on them – Figures 1C and 4B/C. Why was this not performed?  
  3. In examining the Western blots, both in the article and the supplementary, there appears to be a bit of noise in the blots. How can the authors be sure the bands analysed are the target protein? What kind of validation studies were performed on the antibodies?
  4. What was the source of the primers? Were any checks performed to ensure the primers met the MQIE guidelines recommended performance criteria?
  5. In many figures, the gene expression of several proteins is examined, but there are sometimes no data on how this impacts at the protein level. Why did the authors not for example perform Western analyses on KCC2 and NKCC1 in Figure 5?
  6. It would be useful if the magnification was given either in the figure legend on in the figures themselves.

Author Response

Reviewer 1

The authors present an interesting study examining the influence of a particular mutation in the SCN1A gene in the context of incidence of seizures. Specifically, the authors utilize samples obtained from an individual presenting with the c.434T>C mutation to produce induced neuronal cultures expressing the mutation for in vitro modeling. Thereafter, with the use of a control cell type obtained from a healthy individual, the authors profile the characteristics of the ‘disease’-type neurons; measuring the gene profile of particular targets, the expression and localisation of pertinent proteins, and the measurement of action potentials, to name but a few. Together, the authors demonstrate what differences in the ‘disease’ phenotype may lead to increased incidence of seizures, and furthermore, highlight the potential of this approach to complement existing disease modeling approaches in developing therapies and interventions.

In reviewing the manuscript however, I noted a few things that the authors should assess. The following should be considered when preparing a suitable revision.

  • The manuscript utilizes microscopy on a number of occasions in assessing the expression of particular markers, with estimates provided on the co-expression of markers relative to one another in the samples examined. These data are illustrated in the form of bar graphs (e.g. Figure 2D). It is difficult to give these data weight without knowing more on the sampling strategy used to quantify this. The figure legend suggests 300 counts were performed or 200 counts depending, but how were the fields selected? Why weren’t several preparations prepared such that the consistency of the result was examined? It appears this was performed once owing to the lack of error bars. The authors must provide additional information on these data sets, particularly with regards to the sampling strategy. 

We agree with the reviewer that the sampling strategy used to quantify immunofluorescence and relative bar graphs (e.g. GAD1 in Fig. 2D or CALB1, CALB2, and SST in Fig. 3D-E-F, respectively) are not clearly indicated and explained. In the previous version of our manuscript, the percentage of GAD1, CALB1, CALB2, and SST-positive cells were calculated compared to MAP2+ cells in two independent immunostaining experiments. 

In the revised manuscript, we analyzed separately the data obtained from two independent experiments. In this way, bar graphs are represented with error bars (please see Fig. 2D and Fig. 3D-E-F). Captions were modified accordingly. We have added the description of how the fields were selected for the analysis (please refer to paragraph 2.6 within the Materials and Methods section). 

  1. There are a number of figures throughout which do not appear to have any statistical analyses performed on them – Figures 1C and 4B/C. Why was this not performed? 

We apologize for not having indicated the statistics.  The revised manuscript contains the statistical analysis relative to figures 1C, 4B, and 4C and the caption was modified accordingly. Fig. 4 B and C were modified: a new biological replicate was added as suggested by Reviewer 2. 

  • In examining the Western blots, both in the article and the supplementary, there appears to be a bit of noise in the blots. How can the authors be sure the bands analyzed are the target protein? What kind of validation studies were performed on the antibodies? 

NaV1.1 protein has a molecular weight of about 230kDa. However, there are evidence suggesting that the molecular weight of VGSCs is subjected to variations under post-translational modifications (e.g. glycosylation) (Laedermann et al., 2013, doi: 10.3389/fncel.2013.00137). This can explain the presence of multiple bands in our immunoblots. Moreover, the same pattern of bands is present in both control and diseased idNs with an enhanced intensity in wild-type cells supporting our findings relatively to a reduced expression level in SCN1AM145T cells. 

  • What was the source of the primers? Were any checks performed to ensure the primers met the MQIE guidelines recommended performance criteria?

All the qPCR data present in our work meet the MIQE requirements allowing the critical evaluation of the quality of our results. All experiments were performed in at least three different biological replicates each of which ran in technica triplicates.  regarding the primers’ source, these were designed in our laboratory using software and database freely available and accessible such as Ensembl Genome Browser (http://www.ensembl.org) to select transcript ID of specific genes, and Primer3 Plus software (https://www.bioinformatics.nl) for specific primer design. All primer pairs were double checked by alignment using Primer Blast database (https://www.ncbi.nlm.nih.gov/tools/primer-blast/) and only those that specifically anneal the gene of interest were chosen for qPCR analysis.

  • In many figures, the gene expression of several proteins is examined, but there is sometimes no data on how these impacts at the protein level. Why did the authors not for example perform Western analyses on KCC2 and NKCC1 in Figure 5.

The purpose of our research paper is to establish an in vitro model of genetic febrile seizure (FS) using induced pluripotent stem cells (iPSCs). Our data allowed us to delineate an important cellular maturity in diseased cells compared to healthy control iPSCs, strengthening the validity of iPSCs technology in modeling human disease phenotypes. The reason why the expression of some target genes was not always tested also at the protein level (immunoblot analysis) relies on the fact that understanding the molecular mechanisms by which a delay in cell maturation is the goal of future investigation. Importantly, our future work will be aimed at understanding how KCC2 and NKCC co-transportes affect GABAergic transmission.

  1. It would be useful if the magnification was given either in the figure legend or in the figures themselves. 

Magnification of immunofluorescence images is now added in the legends of figures in which IF are shown.

Reviewer 2 Report

This is a careful indepth study of the effect of p.M145T SCN1A mutation on neuronal maturation using iPSC derived neurons. I have the following critical remarks:

  1. The strong effect of the p.M145T mutation on expression of SCN1A shown in Fig. 4 is not clear to me. Why here only two replicates are shown, while in all other figures three biological replicates were available?
  2. I assume that the shown biological replicates are different differentiation experiments from different iPSC clones (correct?).
  3. Additionally to point 1. Heterozygous missense mutations in ion channels cause usually gain-of-function effects, since transcript levels remain usually unchanged. The authors should provide at least a hypothetical molecular mechanism to explain the observed transcript level changes by this missense mutation leading to LOF effects.
  4. The statistical significance of the transcript changes in Figs. 4B and C should be provided.

Author Response

Reviewer 2. 

This is a careful in-depth study of the effect of p.M145T SCN1A mutation on neuronal maturation using iPSC derived neurons. I have the following critical remarks:

  1. The strong effect of the p.M145T mutation on expression of SCN1A shown in Fig. 4 is not clear to me. Why here only two replicates are shown, while in all other figures three biological replicates were available? 

A third biological replicate in Fig. 4B and Fig. 4C was added. 

  • I assume that the shown biological replicates are different differentiation experiments from different iPSC clones (correct?).

All biological replicates present in this study derive from separate differentiation of iPSCs to idNs (iPSCs cultured in different flasks and differentiated in different times). Although the cells are derived from the same source, we think it is accepted that separate handling procedures and differentiation classifies it as biologically different. 

  1. Additionally to point 1. Heterozygous missense mutations in ion channels usually cause gain-of-function effects, since transcript levels remain usually unchanged. The authors should provide at least a hypothetical molecular mechanism to explain the observed transcript level changes by this missense mutation leading to LOF effects. 

In 2005, Mantegazza and collaborators (PNAS, doi: 10.1073/pnas.0506818102) have described the M145T mutation of a well conserved amino acid in the first transmembrane of domain I of the human Na(v)1.1 channel alpha-subunit as LoF mutation. The mutation leads to a reduction in the Na+ current density and a positive shift of about 15 mV in the voltage-dependent activation of the channel. Additionally, other studies have performed electrophysiological analysis on resected hippocampal tissue from the patient SCN1AM145T demonstrating that M145T mutation mostly affects the GABAergic activity thus lowering the inhibitory tone (Ruffolo et al., 2020), providing additional support of the loss-of-function feature of M145T mutation. Here, the reduced expression level of SCN1A was assessed at both protein and mRNA levels in different biological replicates and are in line with the results obtained in previous studies. Moreover, Chen et al. (doi: 10.1016/j.bbadis.2017.04.018) reported that malate dehydrogenase 2 (MDH2) post-transcriptionally regulates the expression of SCN1A under seizure conditions. The post-transcriptional control of SCN1A expression in a patient carrying the c.434T>C mutation is one of the major goals of future investigation. 

  1. The statistical significance of the transcript changes in Figs. 4B and C should be provided.

In the revised version of our paper, the statistical significance relative to transcripts changes shown in figures 4B and 4C was added.  

Round 2

Reviewer 1 Report

The authors have suitably addressed my comments and the manuscript is much improved. 

Reviewer 2 Report

The authors have adequately addressed my concerns.